# Determinants of COVID-19 vaccine hesitancy and uptake in sub-Saharan Africa: a scoping review

Michael J Deml [1,2] Jennifer Nyawira Githaiga [2]

MJD and JNG contributed equally.

¹Institute of Sociological Research, Department of Sociology, University of Geneva, Geneva, Switzerland
²Division of Social and Behavioural Sciences, Faculty of Health Sciences, School of Public Health and Family Medicine, University of Cape Town, Cape Town, South Africa

**Correspondence to**
Dr Michael J Deml;
michaeljdeml@gmail.com

## ABSTRACT

**Objective** To identify, describe and map the research tools used to measure COVID-19 vaccine hesitancy, refusal, acceptance and access in sub-Saharan Africa (SSA).

**Design** Scoping review.

**Methods** In March 2022, we searched PubMed, Scopus, Web of Science, Cochrane, Academic Search Premier, MEDLINE, Cumulative Index to Nursing and Allied Health Literature, Health Source Nursing, Africa Wide and APA PsychInfo for peer-reviewed literature in English related to COVID-19 vaccine hesitancy, refusal, acceptance and access in SSA. We used the Preferred Reporting Items for Systematic Reviews and Meta-Analyses extension for Scoping Reviews to guide evidence gathering and as a template to present the evidence retrieval process.

**Results** In the studies selected for review (n=72), several measurement tools were used to measure COVID-19 vaccine hesitancy, acceptance and refusal. These measurements were willingness and intent to vaccinate from the perspectives of the general population, special population groups such as mothers, students and staff in academic institutions and healthcare workers and uptake as a proxy for measuring assumed COVID-19 vaccine acceptance. Measurements of access to COVID-19 vaccination were cost and affordability, convenience, distance and time to travel or time waiting for a vaccine and (dis)comfort. Although all studies measured COVID-19 vaccine hesitancy, acceptance and refusal, relatively few studies (n=16, 22.2%) included explicit measurements of access to COVID-19 vaccination.

**Conclusions** Based on the gaps identified in the scoping review, we propose that future research on determinants of COVID-19 vaccination in SSA should further prioritise the inclusion of access-related variables. We recommend the development and use of standardised research tools that can operationalise, measure and disentangle the complex determinants of vaccine uptake in future studies throughout SSA and other low- and middle-income country (LMIC) settings.

## INTRODUCTION

In 2019, the WHO listed *vaccine hesitancy* among 10 threats to global health. Predating the advent of the COVID-19 pandemic, this announcement defined vaccine hesitancy as 'the reluctance or refusal to vaccinate despite the availability of vaccines' and pointed to the complex issues underscoring

## STRENGTHS AND LIMITATIONS OF THIS STUDY

⇒ We followed the Preferred Reporting Items for Systematic Reviews and Meta-Analyses extension for Scoping Reviews guidelines to retrieve peer-reviewed publications in English from 10 databases about COVID-19 vaccine hesitancy, refusal, acceptance and access in sub-Saharan Africa.

⇒ The scoping review was guided assisted by a university librarian with expertise in scoping reviews.

⇒ The scoping review process allowed us synthesise and map current evidence, to provide a broad picture of how relatively few studies have so far have measured issues related to COVID-19 vaccine access, especially in combination with vaccine hesitancy, refusal and acceptance.

⇒ The decision to exclude grey literature (conference proceedings, reports, opinion pieces, commentaries) and non-English language texts in our analysis may have limited the data that was available to us.

why people might not get vaccinated, such as 'complacency, inconvenience in accessing vaccines and lack of confidence'.[1] Social and behavioural health scientists researching vaccine hesitancy and vaccine uptake—whether people get vaccinated or not—have long been working on these questions, with a systematic review from a global perspective arguing that there is no 'universal algorithm' (p. 2155) and that the determinants of vaccine hesitancy are complex, context specific and vary across time, place and vaccine.[2] A fundamental message to adequately understand and address *underimmunisation*, or vaccination rates that do not meet public health targets, is that vaccine hesitancy as a determinant for vaccine uptake needs to be disentangled from other determinants unrelated to people's reluctance to vaccinate. Bedford *et al*, for example, explain how hesitancy can be 'used inaccurately as the explanation for undervaccination in a population when the causes are related to pragmatics, competing priorities, access, or the failure of services or policies' (p. 6656).[3]

Before the COVID-19 pandemic began, research on determinants of vaccine uptake had typically been conducted in high-income countries (HICs) with developed healthcare systems and overall regular and dependable access to vaccination for eligible populations. Much of this research focused on parental vaccine hesitancy and pointed to vaccine refusal in HICs as a privileged parenting practice, noting how parents who refused vaccination counted on having adequate access to medical care should their non-vaccinated or undervaccinated children fall ill from vaccine preventable diseases.[4–6] Other studies from HICs have pointed to some parents' adherence to alternative conceptions of health, complementary medicine and neoliberal parenting practices as factors that influence vaccine hesitancy and vaccine refusal.[7–15] Some studies in these settings have particularly focused on the important roles healthcare professionals play in parents' vaccine decision-making process, citing children's doctors as the most important and trusted source of vaccination information.[15–19]

Comparatively fewer social and behavioural vaccine attitude and uptake studies had been conducted in low- and middle-income countries (LMICs) than in HICs before the COVID-19 pandemic. Such studies tended to focus on lack of education, inequality and access issues, rumours about vaccination and 'non-biomedical' approaches to medicine in these countries as determinants of parents' vaccination decisions.[2 20 21] However, research has been increasing in LMICs, with a particular focus on COVID-19 vaccine attitudes and uptake, both in anticipation of and following the arrival of safe and effective vaccines.

Our focus is sub-Saharan Africa (SSA), where healthcare systems are characterised by three distinctive features: (1) high disease burden, (2) inadequate resources and (3) challenges related to leadership and governance. These three features influence public access to healthcare, including quality of service delivery and how systems respond to mundane events and crises such as epidemic outbreaks. First, SSA healthcare systems are not only strongly affected by a high burden of communicable diseases (eg, HIV, tuberculosis malaria, and diarrheal diseases), non-communicable diseases (eg, heart disease, obesity, diabetes and mental illness) and maternal and child mortality; they also grapple with illnesses arising from climate change and environmental pollution and violence-related injuries both at interpersonal levels and in the context of conflict in fragile states.[22–24] Second, relative to healthcare systems in HICs, SSA healthcare systems are under-resourced with regard to healthcare workers, physical infrastructure and facilities and financial resources with glaring disparities in access to healthcare based on geographical areas (rural vs urban) and socioeconomic strata.[22–24] A recent report on public healthcare in SSA indicated that one in six people live more than 2 hours away from their nearest public hospital, while one in eight people live 1 hour or more away from their closest health centre.[25] Third, challenges related to leadership and governance stem from a combination of historical and political factors in post-independence countries as governments have sought to develop healthcare systems, a period characterised by health reforms, economic instability and subsequent structural adjustment sanctions introduced by international donors such as The World Bank and the International Monetary Fund.[26] Governments' inability to finance healthcare systems has culminated in the growth of public–private partnerships, where governments contract non-state providers to assist in healthcare provision as a means of expanding access to healthcare particularly in marginalised areas.[27]

The COVID-19 pandemic and resulting mitigation measures have exacerbated existing healthcare system challenges, causing significant strain on the limited available resources, which has resulted in poor health outcomes. For instance, strict lockdowns in many SSA countries disrupted provision of non-COVID-19-related health services and led to loss of livelihoods and economic recession[28 29] and low levels of trust in governments' responses to the crisis. Existing socioeconomic disparities have served as barriers in adherence to COVID-19 prevention protocols.[29] An analysis of demographic health surveys in 16 SSA countries revealed that only 33.5% of households had water and soap available to support handwashing practices, with greater access in urban compared with rural areas.[30] For instance, approximately only 25% of South Africans from the poorest quintile and close to 40% of rural citizens had access to soap and water.[30] Similarly, in the context of abject poverty and food insecurity more so during the hard lockdown, the threat of COVID-19 has obscured socioeconomic challenges.[31]

COVID-19 vaccination has featured prominently in discussions globally as well as in SSA. Scholars have noted that whereas such discussions have focused on procurement, supply and financing of vaccines,[32] there is a specific need for engagement with COVID-19 vaccine hesitancy.[28 33] There is a strong need for a nuanced understanding of specific contexts and barriers to COVID-19 vaccine uptake given the existing evidence of varying rates of both vaccine hesitancy and uptake reported in various SSA countries.[33–37] A concise narrative review of global literature reported varying degrees of COVID-19 vaccine hesitancy and acceptance, with high vaccine hesitancy prevalence reported in West and Central Africa.[38] Furthermore, COVID-19 vaccine uptake has lagged considerably in SSA compared with other regions globally.[39] Particularly, given the striking healthcare system disparities between HICs and LMICs, it is essential to understand the underlying determinants of COVID-19 vaccine uptake in a way that allows for a nuanced distinction between uptake as it relates to vaccine attitudes and uptake as it relates to access issues.

## Objective

The primary objective of this scoping review was to identify, describe and map the operationalisation and measurement of COVID-19 vaccine hesitancy, refusal, acceptance and access as these relate to COVID-19 vaccine

uptake in SSA. To our knowledge, limited research has so far attempted to disentangle COVID-19 vaccine attitudes from COVID-19 vaccine access issues as determinants of COVID-19 vaccine uptake in SSA. Therefore, this scoping review seeks to address the following research question: How have researchers operationalised and measured vaccine hesitancy and vaccine access as these variables relate to COVID-19 vaccine uptake in SSA?

## METHODS

This scoping review was informed by Levac *et al*'s[40] version of Arksey and O'Malley's[41] framework for scoping reviews[41] and the scoping review methodology of the Joanna Briggs Institute.[42 43] The Preferred Reporting Items for Systematic Reviews and Meta-Analyses extension (PRISMA) for Scoping Reviews[44 45] was used to guide evidence gathering and as a template to present the evidence retrieval process. There is no review protocol for this scoping review.

### Eligibility criteria
#### Concept
Data sources with information on COVID-19 vaccination, vaccine hesitancy, acceptance, refusal, vaccine access and/or vaccine uptake were included in this review. Studies that did not include any of the listed thematic areas were excluded. Studies authored in English were included, while all non-English articles were excluded.

#### Context
Articles included in this review were either fully or partially SSA based, for example, multicountry studies which included both SSA and non-SSA countries. All studies included were published during the COVID-19 pandemic. Non-SSA studies and pre-COVID-19 studies were excluded.

#### Types of evidence sources
We included peer-reviewed, full-text journal articles comprising primary, empirical studies and reviews. Qualitative, quantitative and/or mixed methods studies were included. The following categories of sources were excluded: abstract only; full text not available; non-peer-reviewed articles; and grey literature (conference proceedings, reports, opinion pieces, commentaries).

### Search strategy and study selection
On 9 March 2022, a research librarian and MJD and JNG collaboratively developed and refined the search strategy to include peer-reviewed articles in English that measured COVID-19 vaccine hesitancy, acceptance, refusal and access in SSA. We excluded grey literature, such as conference proceedings, reports, opinion pieces and commentaries. The search strategy included the following search terms: 'COVID-19' OR 'coronavirus 2019' OR 'SARS-CoV-2' OR 'SARS-2' OR 'severe acute respiratory syndrome coronavirus 2', 'vaccination hesitancy' OR 'vaccine hesitancy' OR 'vaccine refusal' OR 'vaccination

refusal' OR 'vaccine access' OR 'access' OR 'sub-Saharan Africa'. The search term 'sub-Saharan Africa' was used to capture studies conducted within this region. We did not include a date filter as we expected that studies related to COVID-19 would be published during the period of the pandemic. A total of 10 databases were searched for relevant articles: PubMed, Scopus, Web of Science, Cochrane, Academic Search Premier, MEDLINE, Cumulative Index to Nursing and Allied Health Literature (CINAHL), Health Source Nursing, Africa Wide and APA PsychInfo. The search strategy was first used in PubMed and adapted for use in the remaining nine databases and is presented in online supplemental file 1. Articles from all 10 databases were exported to EndNote and duplicates were removed. MJD and JNG manually searched reference lists of articles retrieved from the databases for additional relevant articles. They then screened all articles, removing duplicates undetected by EndNote and articles with content falling outside of the scope of the review.

The process of abstract and title screening, based on the inclusion criteria, commenced with both reviewers piloting CINAHL and APA PsychInfo databases together. Disagreements were discussed and resolved through consensus among authors. The remaining articles and databases were then randomly divided into two and each of the reviewers assigned one subset of articles for independent title and abstract screening. All articles which met the inclusion criteria were selected for full-text review. Some of the articles selected for full review were excluded during full-text review screening.

### Data extraction
MJD and JNG created a data extraction form and independently conducted pilot data extraction on nine randomly selected articles. Following pilot data extraction, the data extraction form was refined to include:
1. General descriptive data, namely, the article reference number in EndNote, year of publication, author(s), publication title, aim, study population and country/countries.
2. Data on methods, such as types of studies, measurement scales and tools used.
3. Sociodemographic details of participants included in the selected studies.
4. Study measurement tools and operationalisation of vaccine hesitancy, vaccine acceptance, vaccine refusal, vaccine access and vaccine uptake.

### Patient and public involvement
As this was a scoping review, patients and the public were not involved in the design, conduct, reporting or dissemination plans of our research.

## RESULTS

A total of 3916 articles were retrieved from database searches in Academic Search Premier (n=558), Africa Wide (n=219), APA PsychInfo (n=64), CINAHL (n=127),

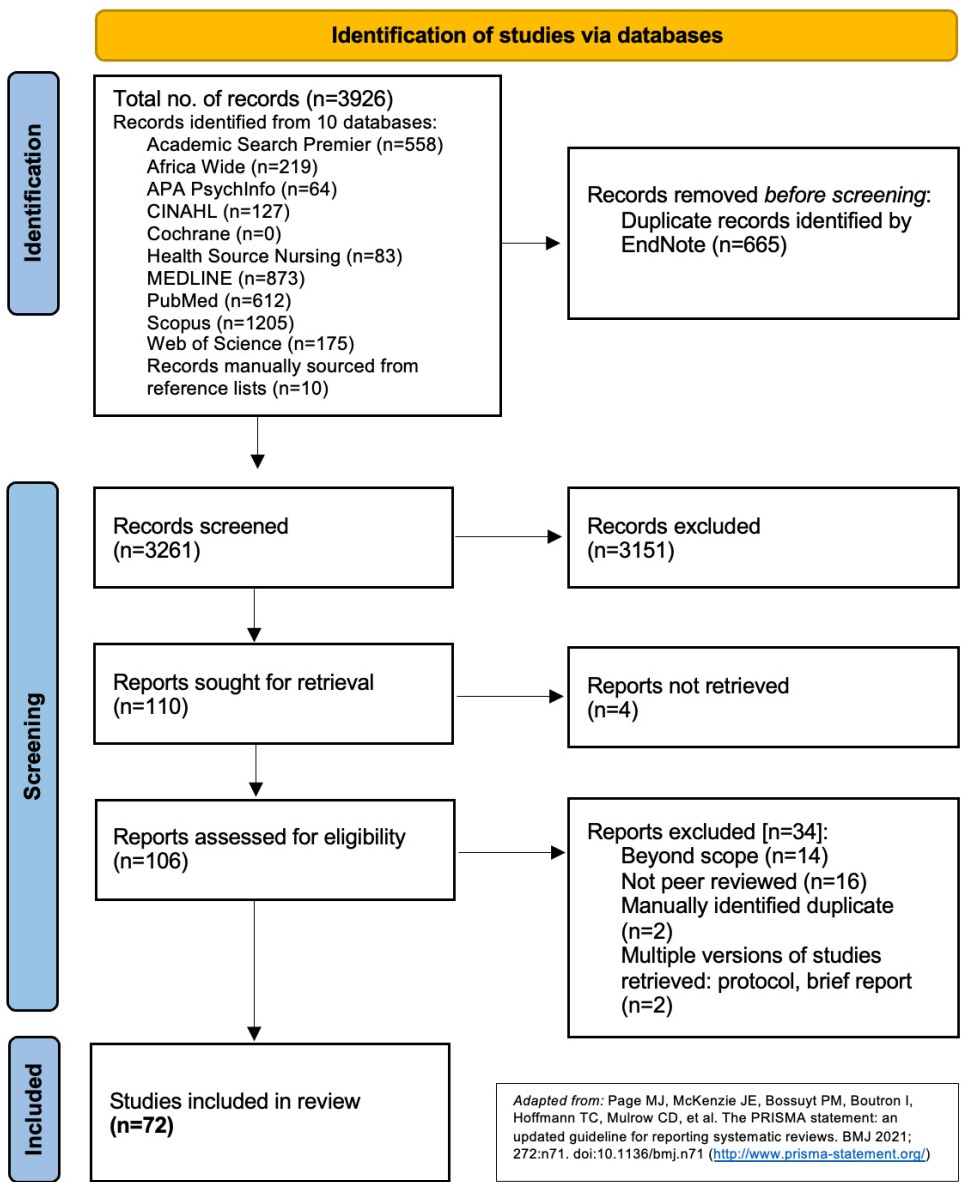

**Figure 1** PRISMA flow diagram. CINAHL, Cumulative Index to Nursing and Allied Health Literature; PRISMA, Preferred Reporting Items for Systematic Reviews and Meta-Analyses.

Cochrane (n=0), Health Source Nursing (n=83), MEDLINE (n=873), PubMed (n=612), Scopus (n=1205) and Web of Science (n=175). Additional articles were manually sourced from reference lists of articles from databases (n=10), yielding a total of 3926 articles. Of these, 665 duplicate records were identified by EndNote and removed. The remaining 3261 articles were screened for eligibility and of these, 3151 articles were excluded. A total of 110 full-text articles were sought for retrieval of which 4 were not available in full text. Of the 106 full-text articles evaluated, 72 studies met the inclusion criteria and were included in this review. The study selection process is captured in a PRISMA flow diagram (figure 1). Online supplemental file 2 includes a list of authors, titles, journal and abstracts of the 72 studies reviewed in the scoping review.

## Characteristics of studies included

The 72 full-text articles reviewed included cross-sectional studies (n=62), systematic reviews (n=4), qualitative studies (n=3), mixed methods studies (n=2) and sentiment analysis (n=1). The articles reviewed comprised data from 58 single country studies and 14 multiple country studies. Not all countries among those listed in the search term for SSA appeared in the 72 articles we reviewed. Of the 58 single country studies, 20 were from Ethiopia, 12 from Nigeria, 6 studies each from Ghana and South Africa, 2 studies each from Kenya, Zimbabwe, Democratic Republic of the Congo and Somalia and 1 study each from Mozambique, Zambia, Togo and Cameroon (table 1). A visual map of all the SSA countries featured in the 72 studies reviewed is presented in figure 2.

| Table 1 | Countries included in reviewed studies |
|---|---|
| **Countries** | **Number of studies** |
| Ethiopia | 20 |
| Nigeria | 12 |
| Ghana | 6 |
| South Africa | 6 |
| Uganda | 2 |
| Kenya | 2 |
| Zimbabwe | 2 |
| Democratic Republic of the Congo | 2 |
| Somalia | 2 |
| Mozambique | 1 |
| Zambia | 1 |
| Togo | 1 |
| Cameroon | 1 |
| Multiple country studies* | 14 |
| Total | 72 |

*Additional sub-Saharan Africa countries included in multiple-country studies were Angola, Benin, Burkina Faso, Cape Verde, Côte d'Ivoire, Gambia, Guinea, Guinea-Bissau, Lesotho, Malawi, Mali, Rwanda, São Tomé and Principe, Senegal, Sierra Leone, Sudan and Tanzania.

Study populations in the 72 reviewed studies comprised general adult populations (n=28), specific adult populations (n=21) including university students, school teachers, chronically ill persons, pregnant women, fully and partially vaccinated adults, mothers, adult caregivers, informal traders and healthcare workers (n=16). Others (n=7) combined two or more populations segments, for instance, school teachers and bank workers in one study and programme personnel, healthcare workers

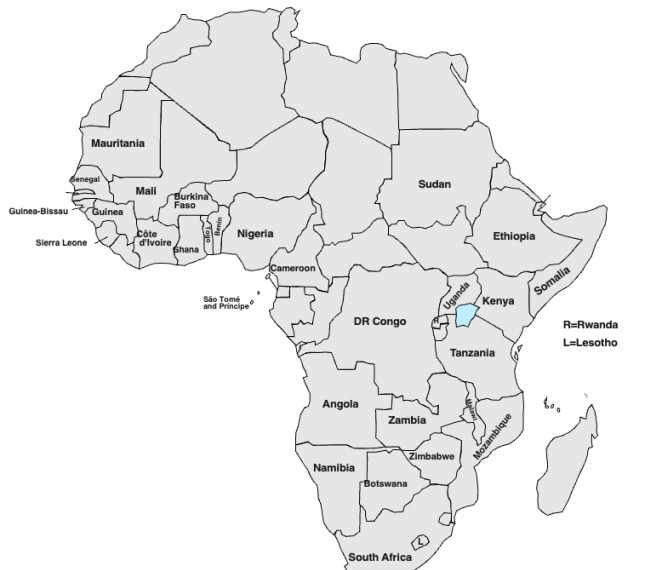

**Figure 2** Map of countries included in reviewed studies.

and community members in another. The main socio-demographic variables captured in the reviewed studies included age, sex, marital status, ethnicity, education, religion, residence, employment status, work category, general health status and, in a few instances, chronic illness status.

## Operationalisation and measurements of vaccine hesitancy, vaccine acceptance and vaccine refusal

We identified different ways researchers operationalised and measured the outcome variables of vaccine hesitancy, vaccine acceptance and vaccine refusal and grouped them into five categories: (1) measurements of willingness to vaccinate (n=32, 44.4%), (2) measurements of intention to vaccinate (n=26, 36.1%), (3) multiple measurements (n=7, 9.7%), (4) uptake measurements (n=4, 5.6%) and (5) qualitative approaches (n=3, 4.2%). We describe these categories in further detail below. We do not provide additional details on operationalisation of the uptake measurement because it is used as a proxy for measuring assumed COVID-19 vaccine acceptance in these studies.[46–49]

### Willingness to vaccinate

The most frequently occurring operationalisation of vaccine hesitancy, acceptance and refusal was willingness to vaccinate against COVID-19 (n=32, 44.4%). Among these studies, 21 included items for which possible responses were 'Yes', 'No' or 'Do not know/Unsure'. For example, Alice Tobin et al[50] asked study respondents, 'Would you be willing to accept a COVID-19 vaccine when one becomes available in the country?' (p. 54).[50] In six studies, researchers included Likert scale responses, such as Addo et al who asked, 'How willing are you to take a COVID-19 vaccine?' (p. 5065).[51] In four studies, researchers added a cost-related condition to the item to measure participants' willingness to pay for a COVID-19 vaccine. Kanyanda et al,[52] for instance, asked participants, 'If an approved vaccine to prevent coronavirus was available right now at no cost, would you agree to be vaccinated?' (p. 2).[52] In one study, researchers asked participants if they would be willing to take the COVID-19 vaccine if it was recommended by a health worker or health agency.[53]

### Intention to vaccinate

We identified intention to vaccinate as a measurement of vaccine hesitancy, acceptance and refusal in 26 (36.1%) of the 72 studies. Among these, 13 included responses for which possible responses were 'Yes', 'No' or 'Do not know/Unsure'. For instance, Abebe et al[54] asked respondents, 'Did you have an intention to accept COVID-19 vaccine if it is available in the future?' (p. 2018).[54] In 10 studies, researchers included Likert scale responses. For example, Wiysonge et al[55] asked study participants to rate their level of agreement on a scale from 1 to 7 (1=strongly disagree, 7=strongly agree) for the statement 'I will take the COVID-19 vaccine when one becomes available'

(p. 3).[55] Researchers included cost-related conditions to measure participants' intention to vaccinate in two studies, including Kassa Mekonnen et al[56] who asked, 'Are you intending to get vaccinated against COVID-19 if available without any cost?' (p. 3).[56]

## Multiple measurements

Seven studies (9.7%) included multiple measurements to operationalise vaccine hesitancy, acceptance and refusal. Chinawa et al measured mothers' willingness to receive the COVID-19 vaccination as well as their willingness to vaccinate their children with the COVID-19 vaccine.[57] Yilma et al asked healthcare workers in Ethiopia if they would get vaccinated if a COVID-19 vaccine was available and proven safe and effective, and if they would recommend their patients to get vaccinated for COVID-19.[58] Sallam[33] conducted a concise systematic review of vaccine acceptance rates and classified acceptance by considering intention to accept, likelihood of vaccination, willingness to accept a vaccine, endorsement of Oxford Scale[59] and level of agreement with vaccination acceptance. In a prevaccination rollout survey in Ghana, Alhassan et al 2021[60] measured respondents' willingness to participate in a COVID-19 vaccine trial and their willingness to take the vaccine. The three remaining studies used multiple items to operationalise vaccine sentiment[61] and vaccine acceptance[62 63] but did not explicitly describe the procedure in full detail.

## Qualitative approaches

Three studies (4.7%) employed qualitative approaches. Wonodi et al conducted focus group discussions and key informant interviews to elicit and thematically analyse COVID-19 vaccine conspiracy theories and misinformation, which they contended may result in 'highly disruptive vaccine hesitancy and refusal' (p. 2115).[64] Shiferie et al used WHO's Strategic Advisory Group of Experts on Immunization (SAGE) definition of vaccine hesitancy ('delay in acceptance or refusal of vaccination despite availability of vaccination services' (p. 4163[65])) in their analysis of 20 qualitative interviews with healthcare providers.[66] In their analysis of documentary, social media and policy analysis, participant observation, ethnography involving informal interviews and observations, Leach et al 2022[64] used the Vaccine Anxieties Framework[20] and argued that it allows for 'exploration of who, in which contexts, really does want COVID-19 vaccines, and may be worried about not getting them' (p. 2).

## Operationalisations and measurements of access to COVID-19 vaccination

Out of the 72 reviewed studies, 16 (22.2%) included operationalisations of access issues related to obtaining COVID-19 vaccines. We grouped these operationalisations into five categories: (1) measurements of cost and affordability (n=13, 18.1%), (2) measurements of convenience (n=6, 8.3%), (3) measurements of distance or time to travel or time waiting for a vaccine (n=3, 4.2%),

(4) measurements of comfort (n=1, 1.4%) and (5) qualitative approaches (n=1, 1.4%). Of these 16 studies, 9 included measurements of access from more than 1 of these categories.

## Cost and affordability

For the measurements of cost and affordability category, 8 of the 13 studies included only a cost and affordability measurement as an operationalisation of access. The other five included additional access items that fell into the other categories. Some of these cost and affordability questions were the same questions discussed above in the willingness and intention to vaccinate measurements (ie, 'If an approved vaccine to prevent coronavirus was available right now at no cost, would you agree to be vaccinated' (p. 2)[52]). Others asked questions about preferences for free vaccines or asked participants to indicate how much they would be willing to pay for a vaccine. Anjorin et al, for example, asked participants to indicate their level of agreement with the following statement: 'If there is a vaccine available for coronavirus, I believe it should be free' (cited from Anjorin et al, S1 File, p.4).[67] The same researchers provided the statement, 'I consider [—] to be a reasonable price range for the coronavirus vaccine' to participants and asked them to choose from the following options: (1) $1–3, (2) $4–6, (3) $7–9 and (4) ≥$10 (cited from Anjorin et al, S1 File, p.4).

## Convenience

We found measurements of convenience as they relate to COVID-19 vaccine acquisition in six studies. Three of these studies asked respondents about general difficulty in accessing vaccination sites. For instance, Orangi et al asked if participants found vaccination sites hard to access.[68] Katoto et al conducted a study in South Africa and asked respondents about their ability to access to the online vaccine registration platform, which has implications for vaccine access pragmatics.[69] Wiysonge et al asked participants about their level of agreement with the statement, 'For me, it is inconvenient to receive vaccinations against COVID-19' (p. 3).[55] Anjorin et al asked respondents if they would prefer community workers to come to their house or place of work to give the coronavirus vaccine, as opposed to going to a health centre (cited from Anjorin et al, S1 File, p.4).[67]

## Distance and time to travel or time waiting for a vaccine

Three studies in total included items about distance/time to travel or time waiting for a vaccine. Davis et al explain how 'self-reported distance and waiting times in queue were used as a means of measuring perceived access to vaccine' (p. 12).[62] Alice Tobin et al asked respondents if they were willing to travel for more than 1 hour to get a COVID-19 vaccine.[50] Anjorin et al ask two similar questions about typical travel time to nearest health centres and the amount of time participants would be willing to travel to get the coronavirus vaccine.[67]

## Comfort

One study included a question about comfort as a measurement related to COVID-19 vaccine access. Wiysonge *et al* asked participants about their level of agreement with the statement 'Visiting the vaccination clinic will make me feel uncomfortable; this will keep me from getting vaccinated against COVID-19' (p. 3).[55]

## Qualitative approaches

One of the 72 studies included qualitative approaches to operationalise COVID-19 vaccine access. In this study, Leach *et al* posit a link between vaccine-related anxiety and access to vaccines based on the availability and equity of resources and observe how the issue of vaccine access is more intricate and unpredictable than presented in ongoing global debates about vaccination.[70]

## Identified gaps

The results of this scoping review allowed us to identify gaps in the current research on COVID-19 vaccine hesitancy, vaccine acceptance, vaccine refusal and vaccine access in SSA. We identified three main gaps in this research: (1) a small proportion of studies investigating issues of COVID-19 vaccine access as a determinant of vaccine uptake, (2) a lack of standardised, homogeneous approaches to measuring COVID-19 vaccine hesitancy, vaccine acceptance, vaccine refusal and vaccine access and (3) a lack of country-wide representative studies.

A major gap in the literature became apparent when we considered the surprisingly low number of studies (n=16, 22.2%) that included study items aimed at measuring COVID-19 vaccine access. Almost all of these studies included measurements related to cost and affordability of the vaccine, while very few considered obstacles individuals might face as barriers to receiving a COVID-19 vaccine, such as accessing online vaccine registration platforms, travel distance and waiting times to reach vaccination centres or sites and comfort when visiting vaccination clinics.

We also identified heterogeneous research approaches to measuring vaccine hesitancy, acceptance, refusal and uptake. The variety of approaches used by researchers throughout SSA likely reflects the difficulties involved when attempting to operationalise admittedly complex phenomena. Similarly, the use of a variety of tools and measurements renders cross-country comparison challenging.

Results of this scoping review also showed that there were relatively few studies that provided country-wide, representative results. Rather, many studies were institution-based, convenience samples or included non-random samples via questionnaires conducted online.

## DISCUSSION

Research on COVID-19 vaccine hesitancy, acceptance, refusal and uptake in SSA has been heterogeneous in terms of study sample populations, study settings, study designs and measurement tools. This is not surprising given the fast-changing nature of the COVID-19 pandemic. This was also coupled with the urgent and complex mass vaccination rollout efforts designed to immunise the highest number of eligible individuals possible in resource-limited settings. This scoping review described the diversity of this research and showed a considerable amount of research about COVID-19 vaccine hesitancy, acceptance and refusal. Nonetheless, few of these studies have included explicit measurements of access to COVID-19 vaccination.

Some of the above-mentioned gaps are likely a result, in part, of the reviewed studies' overall limited engagement with and use of research tools and measurement scales which predated the COVID-19 pandemic. Further consideration of these sources in the study design process would likely have allowed researchers to address some of these gaps.

Several studies did nonetheless adapt literature and models pre-existing the COVID-19 pandemic for use in the context of COVID-19 vaccination. Anjorin *et al*,[67] for example, referenced a 2014 WHO's SAGE report,[71] describing the '3Cs model' which includes the concepts of confidence, complacency and convenience. Anjorin *et al*'s utilisation of the 3Cs model likely prompted them to include items designed to measure variables related to COVID-19 vaccine access, notably through use of the concept convenience.

Wiysonge *et al*[55] explicitly stated that their study questionnaire was informed by the 5C scale from Betsch *et al*,[72] which is an adaptation of SAGE's 3Cs model. The 5C scale measures five psychological antecedents of vaccination: confidence, complacency, constraints, rational calculations of pros and cons and collective responsibility. Wiysonge *et al*'s use of the 5C scale allowed the researchers to include questions related to intention to vaccinate against COVID-19, convenience of getting vaccinated and comfort in going to vaccination clinics. It is notable that there is also now a 7C model that additionally includes measurements of compliance and conspiracy.[73]

Katoto *et al* used the WHO and UNICEF's Behavioural Social Drivers of COVID-19 vaccination (BeSD) tool[74] to inform the development of data collection tools for their study. The BeSD tool assesses four domains related to vaccine uptake: (1) what people think and feel about vaccinations, (2) social processes promoting or hindering vaccination, (3) individual motivations to seek vaccination and (4) practical elements involved in obtaining and getting immunisation. Katoto *et al* noted that the BeSD has limited use in LMICs, which prompted the research team to extensively adapt the tool for the South African context. Nonetheless, use of the BeSD tool in the study design facilitated the inclusion of an item related to practical elements involved in obtaining and getting immunisation: access to the online vaccine registration platform.

Regarding our study objective to identify, describe and map research measurement tools COVID-19 vaccine hesitancy, refusal and acceptance and COVID-19 vaccine access in SSA, our results show that all 72 reviewed studies

included measurements of vaccine hesitancy, refusal and/or acceptance. However, only 16 (22%) studies included at least one measurement of COVID-19 vaccine access. This important finding aligns with a trend developed during the COVID-19 pandemic whereby journalists, governments, policymakers and researchers have increasingly used 'vaccine hesitancy' as an explanation for why so many people remain unvaccinated, even in contexts where there are inadequate vaccine supplies or difficulties accessing vaccination services.[75] In effect, Attwell *et al* observed that papers mentioning 'vaccine' or 'vaccination' in the title, as well as 'hesitancy', increased from 3.3% in 2019 to 8.31% in 2021 (p. 574). These authors argue that this increased focus on vaccine hesitancy 'lets governments off the hook' by centring 'too much of the responsibility for the success (or not) of a vaccination programme on individuals' (ibid).

Our search strategy has limitations. Our decision not to include grey literature, such as conference proceedings, reports, opinion pieces and commentaries, and non-English texts in our review may have limited the available data. There may have been other measurements of vaccine hesitancy, refusal or acceptance around COVID-19 vaccine in SSA reported in the excluded literature and in languages other than English. It should also be noted that the search was conducted in March 2022, so there are likely additional publications that have become available since we conducted the scoping review.

Future research on COVID-19 vaccination in SSA, and other LMIC settings for that matter, needs to prioritise the inclusion of access-related measurements. Inclusion of access variables in future research will add an essential factor to the complex equation around determinants of vaccine uptake. More importantly, its inclusion will fill a current empirical blind spot around COVID-19 vaccine research in SSA whose results have potential to provide insights into concrete, pragmatic and actionable changes designed to make it easier for individuals to obtain COVID-19 vaccines.

## CONCLUSION

This scoping review described the heterogeneity in 72 reviewed studies about COVID-19 vaccine hesitancy, acceptance, refusal and access in SSA. This heterogeneity was apparent in the distribution of countries included, the study designs, sample populations, measurements of vaccine hesitancy, acceptance, refusal, uptake and access. Particularly, we have identified an important empirical blind spot in the literature regarding measurements of vaccine access. Future measurement tools can find inspiration from pre-existing scales, tools and models used for the study of the determinants of vaccine uptake,[65 71 72 74] as was demonstrated in several of the 72 studies reviewed in this scoping review. These research tools should nonetheless be adaptable to capture the local realities specific to the diverse contexts represented in SSA and other LMICs.

**Acknowledgements** The authors would like to thank Nahmla Madini who provided invaluable assistance and support through her role as a librarian at the University of Cape Town's Bongani Mayosi Health Sciences Library. Finally, the authors would like to thank Vladimir Jolidon and Lucia Knight for their valuable inputs for revisions of the scoping review.

**Contributors** Both authors worked together in sourcing funding for this project, conceptualising and designing the study, data collection, analysis, preparation, review and editing of the manuscript and read and approved the final version of the manuscript for submission. MJD accepts full responsibility as the guarantor of this work.

**Funding** This work was supported by the Leading House Africa Research Partnership Grant I and by the Swiss National Science Foundation's sponsorship of MJD's Early Postdoc.Mobility Fellowship (Grant: 200180).

**ORCID iDs**
Michael J Deml http://orcid.org/0000-0003-2224-8173
Jennifer Nyawira Githaiga http://orcid.org/0000-0002-4511-9393

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
