## [Reviewer comments · BMJ Open]

ARTICLE DETAILS

TITLE (PROVISIONAL)	Determinants of COVID-19 vaccine hesitancy and uptake in sub-Saharan Africa: A scoping review
AUTHORS	Deml, Michael; Githaiga, Jennifer

VERSION 1 – REVIEW

REVIEWER	Sallam, Malik The University of Jordan
REVIEW RETURNED	05-Aug-2022

GENERAL COMMENTS	Thanks for the invitation to review the current manuscript. In this scoping review, Michael J. Deml and Jennifer Nyawira Githaiga tackled an important and timely topic; namely, the gaps in knowledge regarding COVID-19 vaccine acceptance/uptake in sub-Saharan Africa. The importance of such a review is the previous and current evidence showing that COVID-19 vaccine hesitancy is widely prevalent in the countries of the region besides the low vaccine uptake in sub-Saharan Africa (based on a recent review: https://doi.org/10.2147/JMDH.S347669 and the NY times website Tracking Coronavirus Vaccinations Around the World: https://www.nytimes.com/interactive/2021/world/covid-vaccinations-tracker.html) Overall, I enjoyed reading the review and learnt a lot since the review is well written with robust methodology. The Introduction provided a proper overview of the review topic. The discussion was concise and relevant and identified the gaps in knowledge based on the available literature on the study topic. I have the following minor comments and suggestions that hopefully can help the authors to improve the final manuscript: 1. The Introduction is comprehensive with proper citation of the relevant literature. However, I might suggest adding a short paragraph highlighting the high prevalence of COVID-19 vaccine hesitancy especially in West/Central Africa citing the following reference: https://doi.org/10.2147%2FJMDH.S347669; besides highlighting the low rates of COVID-19 vaccine uptake in a majority of sub-Saharan Africa countries citing the following website: https://www.nytimes.com/interactive/2021/world/covid-vaccinations-tracker.html2. Additionally, the Introduction can benefit from citing the recent 7C determinants that measure vaccination readiness (I understand the authors focused on convenience, confidence and complacency; however, they can benefit from adding the other factors in the Introduction namely collective responsibility, calculation, conspiracy and compliance) citing the following paper: https://doi.org/10.1027/1015-5759/a0006633. The Methods were described in sufficient details. However, I
---

	might suggest adding a clear paragraph showing the inclusion and exclusion criteria for the studies in this review. In addition, I suggest defining clearly what countries were included in the definition of sub-Saharan Africa. I understand the authors clarified this issue in Figure 2 “Countries Featured in Reviewed Studies”; however, some countries can be classified as SSA countries and were not included in the review (e.g. Chad, Niger, South Sudan, the Central African Republic). Therefore, the authors are encouraged to be clear regarding the countries included in this review. 4. In the Results section, the authors stated that “A total of 3916 articles were retrieved from database searches”. However, in Figure 1 “PRISMA Flow Diagram” the total number of retrieved records was 3926. Please resolve this discrepancy. 5. One final note regarding the manuscript title. I think it can be improved since the scoping review focused on other factors besides the determinants of COVID-19 vaccine uptake.
--	--

REVIEWER	Gautam , Rajesh K. Dr. Harisingh Gour Central University
REVIEW RETURNED	15-Aug-2022

GENERAL COMMENTS	Measuring determinants of COVID-19 vaccine uptake in sub-Saharan Africa: A scoping review I appreciate the efforts made by authors to represent SSA region as it is least reported. The issue opted is also very relevant i.e. COVID-19 vaccination, hesitancy, intention, willingness etc. The manuscript has many merits to be published, still it should be improved. Through revision of language is required. The title should be revised as “Determinants of COVID-19 vaccination in sub-Saharan Africa: A scoping review”. Page wise and line wise comments are given below:  1. Page 3, L10-15: It is objectives under Abstract. It requires to be rewritten. 2. Page 3, L36-41: This section is quite vague and requires rephrasing. 3. Page 4, L10-22: Conclusion under abstract need to be rewritten based on the results. 4. Page 5, L 36: There is repetition of word 'that' 5. Page 7, L3: It start with "we here focus.....". It should be written in third person or it should be rephrased as "This study focused on....." 6. Page 8, L38-49: This section needs to be re-written. 7. Page 9, L36-43: Objective should be placed with Introduction in spite of Methods. 10. Page 10, L35-47: This section needs to be re-written. 11. Page 11, L3-12: This section needs to be re-written. 12. Page 12, L17-45: Including PRISMA diagram should be placed under METHODS. Results should be started with findings rather than describing the method or methodological details. 13. Page 14, L20 onward should be beginning of the RESULT section. Before that is parts of METHODS. Further, the L20 should not start with 'we'. Where ever 'we' is used should be replaced by third person. 14. Page 12, L47: Characteristics of studies included. This whole section can be shifted under METHODS. 15. Discussion and Conclusion section also need to be rewritten. 16. Page 32: Table. In-spite of Table of Authors and Title only; The Table should include name of Author, Year, Region/country, Method and Main Findings.
--

VERSION 1 – AUTHOR RESPONSE

Reviewer: 1

Dr. Malik Sallam, The University of Jordan

Comments to the Author:

Thanks for the invitation to review the current manuscript.

In this scoping review, Michael J. Deml and Jennifer Nyawira Githaiga tackled an important and timely topic; namely, the gaps in knowledge regarding COVID-19 vaccine acceptance/uptake in sub-Saharan Africa. The importance of such a review is the previous and current evidence showing that COVID-19 vaccine hesitancy is widely prevalent in the countries of the region besides the low vaccine uptake in sub-Saharan Africa (based on a recent review: <https://doi.org/10.2147/JMDH.S347669> and the NY times website Tracking Coronavirus Vaccinations Around the World:

<https://www.nytimes.com/interactive/2021/world/covid-vaccinations-tracker.html>)

Overall, I enjoyed reading the review and learnt a lot since the review is well written with robust methodology. The Introduction provided a proper overview of the review topic. The discussion was concise and relevant and identified the gaps in knowledge based on the available literature on the study topic.

Response: We thank the reviewer for the constructive appreciation and feedback for the manuscript.

I have the following minor comments and suggestions that hopefully can help the authors to improve the final manuscript:

1. The Introduction is comprehensive with proper citation of the relevant literature. However, I might suggest adding a short paragraph highlighting the high prevalence of COVID-19 vaccine hesitancy especially in West/Central Africa citing the following reference:

<https://doi.org/10.2147%2FJMDH.S347669>; besides highlighting the low rates of COVID-19 vaccine uptake in a majority of sub-Saharan Africa countries citing the following website:

<https://www.nytimes.com/interactive/2021/world/covid-vaccinations-tracker.html>

Response: The recommended readings and sources are duly appreciated. We have added additional sentences to the Introduction with this information.

2. Additionally, the Introduction can benefit from citing the recent 7C determinants that measure vaccination readiness (I understand the authors focused on convenience, confidence and complacency; however, they can benefit from adding the other factors in the Introduction namely collective responsibility, calculation, conspiracy and compliance) citing the following paper:

<https://doi.org/10.1027/1015-5759/a000663>

Response: We thank the reviewer for this recommendation and the reference to the 7C determinant scale. We have chosen to include a reference to it in the Conclusion section when we explicitly discuss the previous 5C model.

3. The Methods were described in sufficient details. However, I might suggest adding a clear paragraph showing the inclusion and exclusion criteria for the studies in this review.

Response: We have clarified the inclusion and exclusion criteria in the Methods section.

In addition, I suggest defining clearly what countries were included in the definition of sub-Saharan Africa. I understand the authors clarified this issue in Figure 2 “Countries Featured in Reviewed

Studies”; however, some countries can be classified as SSA countries and were not included in the review (e.g. Chad, Niger, South Sudan, the Central African Republic). Therefore, the authors are encouraged to be clear regarding the countries included in this review.

Response: We thank the reviewer for pointing to this issue. We have clarified in the Methods section that the search term ‘sub-Saharan Africa’ was designed to capture all countries within the MESH term for this region. We have included the list of these countries in Supplementary file 1. We have also clarified in the results that, despite including this search term, not all sub-Saharan Africa countries were represented in the 72 selected studies.

4. In the Results section, the authors stated that “A total of 3916 articles were retrieved from database searches”. However, in Figure 1 “PRISMA Flow Diagram” the total number of retrieved records was 3926. Please resolve this discrepancy.

Response: We understand the confusion. We first retrieved 3916 articles from the databases and manually sourced 10 additional relevant sources from reference/bibliography lists of retrieved articles, resulting in a total of 3926 records considered for this review. This is explained in the 2nd sentence of the 1st paragraph of the results.

5. One final note regarding the manuscript title. I think it can be improved since the scoping review focused on other factors besides the determinants of COVID-19 vaccine uptake.

Response: We have revised the title to “Determinants of COVID-19 vaccine hesitancy and uptake in sub-Saharan Africa: A scoping review”

Reviewer: 2

Rajesh K. Gautam , Dr. Harisingh Gour Central University

Comments to the Author:

Measuring determinants of COVID-19 vaccine uptake in sub-Saharan Africa: A scoping review
I appreciate the efforts made by authors to represent SSA region as it is least reported. The issue opted is also very relevant i.e. COVID-19 vaccination, hesitancy, intention, willingness etc.

Response: We thank the reviewer for the appreciation of the importance of the topic and the manuscript.

The manuscript has many merits to be published, still it should be improved. Through revision of language is required.

Response: We have thoroughly proofread the manuscript for language, with changes made in track change throughout.

The title should be revised as “Determinants of COVID-19 vaccination in sub-Saharan Africa: A scoping review”.

Response: We have revised the title to “Determinants of COVID-19 vaccine hesitancy and uptake in sub-Saharan Africa: A scoping review”

Page wise and line wise comments are given below:

1. Page 3, L10-15: It is objectives under Abstract. It requires to be rewritten.

Response: We have shortened the objective and clarified the sentence.

2. Page 3, L36-41: This section is quite vague and requires rephrasing.

Response: We appreciate the note and have reworded this section for clarity.

3. Page 4, L10-22: Conclusion under abstract need to be rewritten based on the results.

Response: We have rewritten this section to highlight that the scoping review identified an important gap in the literature around access-related research measurements.

4. Page 5, L 36: There is repetition of word 'that'

Response: We thank the reviewer for catching this and have removed the repeated word.

5. Page 7, L3: It start with "we here focus.....". It should be written in third person or it should be rephrased as "This study focused on....."

Response: We respectfully disagree and argue that this is a stylistic choice. Our preference is to use "we" to avoid ambiguity. We would be happy to re-consider upon editorial recommendation.

6. Page 8, L38-49: This section needs to be re-written.

Response: We have reworded one of these sentences for clarity and have added additional information in response to another reviewer's request.

7. Page 9, L36-43: Objective should be placed with Introduction in spite of Methods.

Response: We have moved this section to the Introduction section.

10. Page 10, L35-47: This section needs to be re-written.

Response: Thank you for pointing this out. Upon re-reading, we noticed a typo and have removed the 2 words "from the" and have clarified the inclusion and exclusion criteria.

11. Page 11, L3-12: This section needs to be re-written.

Response: Thank you for pointing this out. Upon re-reading, we noticed a typo and have rephrased as "search strategy was first used in." We have also referenced Supplementary file 1.

12. Page 12, L17-45: Including PRISMA diagram should be placed under METHODS. Results should be started with findings rather than describing the method or methodological details.

Response: We have followed the PRISMA-ScR checklist from Tricco et al. (2018) (<https://www.equator-network.org/reporting-guidelines/prisma-scr/>), which asks for this information under the Results section.

13. Page 14, L20 onward should be beginning of the RESULT section. Before that is parts of METHODS.

Response: We have followed the PRISMA-ScR checklist from Tricco et al. (2018) (<https://www.equator-network.org/reporting-guidelines/prisma-scr/>), which asks for this information under the Results section.

Further, the L20 should not start with 'we'. Where ever 'we' is used should be replaced by third person.

Response: We respectfully disagree and argue that this is a stylistic choice. Our preference is to use “we” to avoid ambiguity. We would be happy to re-consider upon editorial recommendation.

14. Page 12, L47: Characteristics of studies included. This whole section can be shifted under METHODS.

Response: We have followed the PRISMA-ScR checklist from Tricco et al. (2018) (<https://www.equator-network.org/reporting-guidelines/prisma-scr/>), which asks for this information under the Results section.

15. Discussion and Conclusion section also need to be rewritten.

Response: We have proofread and revised the Discussion and Conclusion sections for clarity.

16. Page 32: Table. In-spite of Table of Authors and Title only; The Table should include name of Author, Year, Region/country, Method and Main Findings.

Response: This is a great suggestion. We have updated “Supplementary file 2” to include the Author, Year, Journal, and Abstract, which includes the information requested by Reviewer 2.

VERSION 2 – REVIEW

REVIEWER	Sallam, Malik The University of Jordan
REVIEW RETURNED	25-Aug-2022
GENERAL COMMENTS	Thanks for addressing my previous comments thouroughly.
REVIEWER	Gautam , Rajesh K. Dr. Harisingh Gour Central University
REVIEW RETURNED	11-Sep-2022
GENERAL COMMENTS	Authors have made significant improvement in the manuscript, still some to the earlier comments were not addressed. It is suggested to improve the manuscript as suggested earlier. The Table is again vague as they copied and pasted abstracts of the studies included. Inspite of just giving a brief of findings. The Table is not acceptable at all in its present form. Some of important and recent literature is also missing e.g. Limbu, Y.B.; Gautam, R.K.; Pham, L. The Health Belief Model Applied to COVID-19 Vaccine Hesitancy: A Systematic Review. Vaccines 2022, 10, 973. https://doi.org/10.3390/vaccines10060973

VERSION 2 – AUTHOR RESPONSE

Reviewer: 1

Dr. Malik Sallam, The University of Jordan

Comments to the Author:

Thanks for addressing my previous comments thoroughly.

With my best wishes

Response: We thank the reviewer for the feedback on the manuscript.

Reviewer: 2

Rajesh K. Gautam , Dr. Harisingh Gour Central University

Authors have made significant improvement in the manuscript, still some to the earlier comments were not addressed. It is suggested to improve the manuscript as suggested earlier.

Response: We are not sure we understand specifically where Reviewer 2 is suggesting that we make changes. We have included below our point-by-point responses to the reviewer's original requests for revisions. We are happy to address any specific concerns.

The Table is again vague as they copied and pasted abstracts of the studies included. In spite of just giving a brief of findings. The Table is not acceptable at all in its present form.

Response: We respectfully disagree and argue that the Reviewer's request to synthesize and summarize the reviewed studies' findings goes beyond the scope and objectives of our scoping review. A synthesis of the studies' findings is better aligned with a systematic review of literature and not with a scoping review. The information Reviewer 2 has requested can be found in the updated Table of our most recent submission. We request that an editorial decision be made on this issue by the journal editor.

We here include the objectives of the study and research questions our scoping review sought to answer:

“The primary objective of this scoping review was to identify, describe and map the operationalization and measurement of COVID-19 vaccine hesitancy, refusal, acceptance and access as these relate to COVID-19 vaccine uptake in SSA. To our knowledge, limited research has so far attempted to disentangle COVID-19 vaccine attitudes from COVID-19 vaccine access issues as determinants of COVID-19 vaccine uptake in SSA. Therefore, this scoping review seeks to address the following research question: How have researchers operationalized and measured vaccine hesitancy and vaccine access as these variables relate to COVID-19 vaccine uptake in sub-Saharan Africa?”

Some of important and recent literature is also missing e.g. Limbu, Y.B.; Gautam, R.K.; Pham, L. The Health Belief Model Applied to COVID-19 Vaccine Hesitancy: A Systematic Review. *Vaccines* 2022, 10, 973. <https://doi.org/10.3390/vaccines10060973>

Response: We thank the reviewer for pointing our attention to his research but would kindly point out that his article was published on June 18, 2022. Our scoping review was conducted on literature published before March 2022. Since our scoping review results did not focus on the health belief model, nor did this article fit within our scoping review's date eligibility criteria, we do not find it appropriate to cite this article as it did not inform our conceptualization of the scoping review.

Reviewer 2's comments to previous the original submission and our original rebuttals are included below.

Comments to the Author:

Measuring determinants of COVID-19 vaccine uptake in sub-Saharan Africa: A scoping review

I appreciate the efforts made by authors to represent SSA region as it is least reported. The issue opted is also very relevant i.e. COVID-19 vaccination, hesitancy, intention, willingness etc.

Response: We thank the reviewer for the appreciation of the importance of the topic and the manuscript.

The manuscript has many merits to be published, still it should be improved. Through revision of language is required.

Response: We have thoroughly proofread the manuscript for language, with changes made in track change throughout.

The title should be revised as “Determinants of COVID-19 vaccination in sub-Saharan Africa: A scoping review”.

Response: We have revised the title to “Determinants of COVID-19 vaccine hesitancy and uptake in sub-Saharan Africa: A scoping review”

Page wise and line wise comments are given below:

1. Page 3, L10-15: It is objectives under Abstract. It requires to be rewritten.

Response: We have shortened the objective and clarified the sentence.

2. Page 3, L36-41: This section is quite vague and requires rephrasing.

Response: We appreciate the note and have reworded this section for clarity.

3. Page 4, L10-22: Conclusion under abstract need to be rewritten based on the results.

Response: We have rewritten this section to highlight that the scoping review identified an important gap in the literature around access-related research measurements.

4. Page 5, L 36: There is repetition of word 'that'

Response: We thank the reviewer for catching this and have removed the repeated word.

5. Page 7, L3: It start with "we here focus.....". It should be written in third person or it should be rephrased as "This study focused on....."

Response: We respectfully disagree and argue that this is a stylistic choice. Our preference is to use “we” to avoid ambiguity. We would be happy to re-consider upon editorial recommendation.

6. Page 8, L38-49: This section needs to be re-written.

Response: We have reworded one of these sentences for clarity and have added additional information in response to another reviewer’s request.

7. Page 9, L36-43: Objective should be placed with Introduction in spite of Methods.

Response: We have moved this section to the Introduction section.

10. Page 10, L35-47: This section needs to be re-written.

Response: Thank you for pointing this out. Upon re-reading, we noticed a typo and have removed the 2 words “from the” and have clarified the inclusion and exclusion criteria.

11. Page 11, L3-12: This section needs to be re-written.

Response: Thank you for pointing this out. Upon re-reading, we noticed a typo and have rephrased as “search strategy was first used in.” We have also referenced Supplementary file 1.

12. Page 12, L17-45: Including PRISMA diagram should be placed under METHODS. Results should be started with findings rather than describing the method or methodological details.

Response: We have followed the PRISMA-ScR checklist from Tricco et al. (2018) (<https://www.equator-network.org/reporting-guidelines/prisma-scr/>), which asks for this information under the Results section.

13. Page 14, L20 onward should be beginning of the RESULT section. Before that is parts of METHODS.

Response: We have followed the PRISMA-ScR checklist from Tricco et al. (2018) (<https://www.equator-network.org/reporting-guidelines/prisma-scr/>), which asks for this information under the Results section.

Further, the L20 should not start with 'we'. Where ever 'we' is used should be replaced by third person.

Response: We respectfully disagree and argue that this is a stylistic choice. Our preference is to use “we” to avoid ambiguity. We would be happy to re-consider upon editorial recommendation.

14. Page 12, L47: Characteristics of studies included. This whole section can be shifted under METHODS.

Response: We have followed the PRISMA-ScR checklist from Tricco et al. (2018) (<https://www.equator-network.org/reporting-guidelines/prisma-scr/>), which asks for this information under the Results section.

15. Discussion and Conclusion section also need to be rewritten.

Response: We have proofread and revised the Discussion and Conclusion sections for clarity.

16. Page 32: Table. In-spite of Table of Authors and Title only; The Table should include name of Author, Year, Region/country, Method and Main Findings.

Response: This is a great suggestion. We have updated “Supplementary file 2” to include the Author, Year, Journal, and Abstract, which includes the information requested by Reviewer 2.